# The Cranial Morphology of the Black-Footed Ferret: A Comparison of Wild and Captive Specimens

**DOI:** 10.3390/ani12192708

**Published:** 2022-10-09

**Authors:** Tyler Antonelli, Carissa L. Leischner, Adam Hartstone-Rose

**Affiliations:** 1Dental Education, University of North Carolina School of Dentistry, Chapel Hill, NC 27599, USA; 2Department of Cell Biology and Anatomy, University of South Carolina School of Medicine, Columbia, SC 29209, USA; 3Department of Biological Sciences, North Carolina State University, Raleigh, NC 27695, USA

**Keywords:** *Mustela nigripes*, captivity, captive breeding, husbandry

## Abstract

**Simple Summary:**

The black-footed ferret, a member of the weasel family originally found throughout much of Midwestern North America, nearly went extinct prior to the 1980s, in part because of ranchers’ persecution of prairie dogs, which make up almost all of the ferret’s natural diet. In the 1980s, the few remaining wild individuals of the species were brought into captivity in an effort to breed enough animals to reintroduce a more stable population back into the wild. While this program was successful in expanding the numbers of animals enough to release animals back into the wild, the diet of the captive animals was so substantially different that it caused dramatic dental issues. In the current study, we examined the skulls of 271 adult ferrets and 53 specimens of two species that are closely related to ferrets and found that the captive ferrets differ substantially in skull shape from the wild individuals and that some of these differences are even more substantial than some of the differences between black-footed ferrets and their relatives. Thus, captivity (probably the captive diet) has a substantial negative effect on not only the oral health (gum disease, cavities, etc.) but also on skull shape as well.

**Abstract:**

The black-footed ferret (*Mustela nigripes*), a North American mustelid species, was once found abundantly throughout the Midwest until the extreme decline in prairie dogs (*Cynomys* spp.), the black-footed ferret’s primary food source, brought the species to near-extinction. Subsequently, the Black-Footed Ferret Recovery Program was created in the 1980s with a goal of bringing all remaining individuals of the species into captivity in order to breed the species back to a sustainable population level for successful reintroduction into the wild. While many components of the ferrets’ health were accounted for while in captivity—especially those affecting fecundity—this study aims to assess the effects that captivity may have had on their cranial morphology, something that has not been widely studied in the species. In a previous study, we showed that the captive ferrets had significant oral health problems, and here we aim to document how the captive diet also affected their skull shape. For this study, 23 cranial measurements were taken on the skulls of 271 adult black-footed ferrets and 53 specimens of two closely related species. Skulls were divided based on sex, species, captivity status and phase of captivity and compared for all measurements using stepwise discriminant analysis as well as principal component analysis derived from the combined variables. We found that there are significant differences between captive and wild specimens, some of which are larger than interspecific variation, and that a diet change in the captive specimens likely helped decrease some of these differences. The results suggest that captivity can cause unnatural cranial development and that diet likely has a major impact on cranial morphology.

## 1. Introdution

The black-footed ferret (*Mustela nigripes*, Mustelidae) is endemic to North America [1]. First described by Audubon and Bachman [2], these nocturnal carnivores were originally found throughout the central portion of the United States and parts of Canada and Mexico [1]. As black-footed ferrets are entirely dependent on prairie dogs (*Cynomys* spp.) for both shelter and food—making up approximately 90% of their diet [1,3,4], they nearly went extinct during the 20th century when prairie dog populations plummeted due to widespread extermination by ranchers. It was estimated that the prairie dog, by 1960, occupied only 2% of its original range [5,6].

By 1964, only one known population black-footed ferrets remained, in South Dakota [7,8]. A captive breeding program was created for this population but ultimately failed, and the species was presumed extinct until 1981, when a previously unknown population was discovered in Wyoming [9,10]. This remnant population of over 100 ferrets was monitored in the wild until 1984 when the U.S. Fish and Wildlife Service, as part of the Black-Footed Ferret Recovery Act, began capturing individuals for a new breeding program [9,11]. In 1987, what is believed to be the last wild ferret was captured, resulting in a final captive population of only 7 males and 11 females [12]. These animals successfully bred and the husbandry proved so successful that beginning in 1991, ferrets were reintroduced into the wild in Wyoming [13,14].

Although captive black-footed ferrets willingly eat ground squirrels (*Spermophilus tridecemlineatus*), rabbits (*Sylvilagus floridanus*), deer mice (*Peromyscus* spp.), and various species of birds and insects [15,16], Vargas and Anderson [16] found that they prefer prairie dog when available.

Initially, to maximize fecundity [13,17], the captive black-footed ferrets were fed a high calorie diet that was predominantly a nutritionally supplemented soft mixture of 60% commercial mink chow pellets and 40% skinned, eviscerated, headless ground rabbit or prairie dog [13]. Each ferret was also fed a freshly killed hamster (*Mesocricetus auratus*) once per week [13,18]. Many of the nutrients present in this “60/40 diet” provided to ferrets held at the Toronto Metro Zoo exceeded quantities found in the natural diet of black-footed ferrets—with specific concerns surrounding the high level of polyunsaturated fatty acids [13]. As a result, the nutritional components of the diet were adjusted during the mid-1990s until the diet was entirely replaced in 2000 by the “Toronto Zoo Diet” [13], a diet (widely fed to captive carnivores throughout North America) made of vitamin-supplemented ground horsemeat that contains “no bones, cartilage, organs, skin or connective tissues” [19]. It was preferred by ferret caretakers because of its convenience (available pre-fortified—deemed a complete diet) and ability to maintain reproductive health [13]. Although the “Toronto Zoo Diet” was a nutritional improvement, black-footed ferrets given this diet have been documented to have had oral health maladies—most notably in excessive periodontal disease and dental calculus [20].

Captive centers strive to provide animals with the most nutritious diets available, yet oftentimes they overlook another key element of diet: mechanical properties. Nutrition is valuable for health and development, but improper mechanical properties can be detrimental. For example, it has been suggested that captivity can cause abnormal effects to cranial shape and development. Anomalies such as cranial thickening in baboons (*Papio* spp.; [21]), and decreased skull sizes in Indian rhinoceros (*Rhinoceros unicornis*; [22]) have been observed in captive animals. More specifically, previous research has found a correlation between captive diets and abnormal cranial morphology and oral health [23,24,25]. The most comprehensive of these studies, conducted by Hartstone-Rose and colleagues [25], found evidence that captivity dramatically alters the shape of the skull of lions (*Panthera leo*) and tigers (*P. tigris*) based on three-dimensional analyses of 43 bony landmarks. Results from that study found that captivity was actually a stronger driving factor (by approximately twice the magnitude) for cranial morphological differences than is sex in these sexually dimorphic species. The study also found that cranial morphological differences in the large felids were due in most part to zygomatic width and biangular width in relation to rostral length measurements [25]—both regions of morphology closely associated with the mandibular adductors. A follow-up study [26] found that lions and tigers with highly distorted crania also have correspondingly poor oral health. Cranial morphology can be better understood when the muscles of mastication are also considered.

Unfortunately, although the masticatory muscles of musteloids have been studied [27,28,29], there have not been any specific dissection-based studies of the muscles of black-footed ferrets. However, investigation of the function of these muscles in other species—particularly carnivorans—can help inform our understanding of their effects on cranial morphology. The masseter, temporalis, and medial pterygoid adduct the mandible and provide power for mastication. These muscles all originate on the cranium and insert onto the posterior portion of the mandible. Decreased area of any of these attachment areas might suggest reduced force generation. Thus, dietary differences (e.g., a softer diet in captivity) may lead to different skull shapes in black-footed ferrets.

While black-footed ferret cranial morphology has not been extensively studied, the taxon has been the subject of numerous other studies, mostly regarding their reproduction and disease prevention [30]. One of the main goals of the Black-Footed Ferret Recovery Act was to restore the population size of the black-footed ferret. To this end, reproduction and disease prevention were rigorously tracked and studied. Studies have investigated genetic diversity [31], courtship [32], age-dependent male fertility [33], canine distemper [34,35,36] and sylvatic plague [37,38,39]. This research was critical to black-footed ferret recovery and reintroduction, but further research into other areas of health, including captivity’s effect on cranial morphology studied here, will serve to continue to improve the conservation efforts of this species.

Our previous study on the effects of captivity on black-footed ferrets found that ferrets in captivity showed higher levels of oral health defects [21]. This study analyzed both captive and wild ferrets for calculus build-up and periodontal disease and determined that both calculus and periodontal disease appeared significantly worse in captive animals than in wild animals. It also found that the specimens accessioned after dietary changes started to be made around 1996 had significantly more calculus and periodontal disease compared to the specimens accessioned before 1996 [21].

The goal of this study is to determine if captivity also affected the *cranial morphology* of black-footed ferrets. Specifically, the present study aims to narrow the statement made by Wisely and colleagues [14] that “decreased [skull] sizes were likely a result of environmental conditions present in captivity” in black-footed ferrets, by trying to establish whether diet composition is a prime reason for the morphological changes brought about in the captive specimens. It is likely that physical dietary properties affect cranial morphology as Hartstone-Rose and colleagues [26] found similar results in captive felids. Mechanical properties of the diet likely influenced the development of masticatory muscles, which then further resulted in the alteration of skull shape and size [26].

### Hypotheses

**Hypothesis 1 (H1).** 
*Black-footed ferrets raised in captivity will have significantly different skull shapes and smaller skull sizes than do wild black-footed ferrets—especially in terms of zygomatic width, a key indicator of masticatory muscle strength—as a result of the unnaturally soft diets presented to captive ferrets.*


**Hypothesis 2 (H2).** 
*Black-footed ferrets bred in captivity following the dietary shift that began in 1996 will have skull measurements less similar to those of wild specimens than do ferrets fed the earlier diet. The gradual change from the “60/40 diet” to the “Toronto Zoo Diet” in the mid-1990s was mainly implemented to increase the nutritional content of the diet, but the mechanical properties of the two diets have only been minimally examined in regard to their effect on cranial morphology. We hypothesize that, because our previous research has shown increased oral disease in specimens accessioned after 1996 [21], the change in diet also resulted in cranial morphology that is less similar to wild specimens than the cranial morphology of captive specimens accessioned prior to 1996.*


**Hypothesis 3 (H3).** 
*The extent of variation between the captive groups and the wild group will be equal to or greater than the extent of variation between the wild black-footed ferret and two closely related species. In comparing the study species to its congeners, points of reference can be established that determine the severity of these cranial changes in comparison to speciation. If modern black-footed ferrets are less like their wild predecessors than they are to entirely separate species, we can begin to see more clearly how drastic these changes to cranial morphology truly are. The European and Steppe polecats (M. putorius and M. eversmanii, respectively) are closely related species to the black-footed ferret but observations have found cranial morphology differences between the species [40]. Based on observations about the morphological effects of captivity in other species [26], our previous findings of dramatically different oral health [21], and our qualitative impression of the distortion of these skulls, we believe that captive black-footed ferrets will be less like wild black-footed ferrets by some metrics than are the shape differences between the wild congeners.*


## 2. Materials and Methods

### 2.1. Sample

Cranial measurements were collected on 242 specimens, 232 from the National Museum of Natural History (specimens cataloged as USNM; Smithsonian, Washington, DC, USA) and 10 specimens from the American Museum of Natural History (AMNH; New York, NY, USA). Of the 242 skulls, 206 are of the focal ferret species, and the rest are its close wild congeners included as outgroup comparative polecat species: 24 *M. putorius* and 12 *M. eversmanii*. The black-footed ferret specimens were divided, as previously described [21], into 4 groups based on time of death and captivity status (Table 1). Only adult skulls with fully erupted secondary dentition were included in the sample.

### 2.2. Skull Measurements

Twenty-three linear measurements were collected (Figure 1 and Table 2) to the hundredth of a millimeter using digital calipers (Mitutoyo) directly onto a spreadsheet.

### 2.3. Analysis

To account for sexual dimorphism, separate analyses were conducted for male and female specimens; specimens of unknown sex were excluded. Analyses were conducted on three different groupings of the specimens: the first compared wild and captive black-footed ferrets for all 23 cranial measurements and significant principal components. The second analysis replicated the first, but divided the ferrets into the 4 groups (wild, early captive, recovery phase I and recovery phase II). The third analysis compared all 4 black-footed ferret groups with the inclusion of the two congener polecat species in a step-wise multivariate discriminate analysis using variables that significantly distinguished the sex-specific samples with an alpha < 0.05). Skull measurements were statistically analyzed (ANOVA) using JMP versions 10.0.2 and 16.0.0 (SAS). For analyses of significant differentiation of individual variables comparing more than two groups, an all pairs, Tukey–Kramer honest significant difference analysis was used to determine variability of means. All results were considered statistically significant for alpha < 0.05.

## 3. Results

### 3.1. Comparison of Wild vs. Captive Black-Footed Ferrets

Wild and captive black-footed ferret skulls are substantially different from each other (Figure 2). Comparing cranial measurements of male wild versus captive black-footed ferrets (Table 3), 12 of the 23 measurements were significantly different (*p*-value < 0.05) with basal length, coronoid height and condyle height all being highly significantly different (*p*-value < 0.001). The captive specimens had a smaller mean than the wild specimens for 10 of the 12 differing measurements, with the two exceptions being maximum height and bicoronion width. Principal Component Analysis of the 23 measurements found that in males, PC1 is driven by overall size (all positive eigenvectors) and accounted for a large portion of the variation in the data (50.25%). This principal component does not significantly separate wild from captive specimens (Table 3). The second principal component, however, does highly significantly separate wild from captive ferrets. Based on the eigenvectors, PC2 (which contributes 9.69% of variation) is driven predominantly by condyle height, post-orbital constriction, and total skull height inversely related to bicoronion width. No other principal components significantly separate the wild and captive male specimens.

In the females, 15 of the 23 measurements significantly differ between wild and captive black-footed ferrets (Table 4). The mean for the wild ferrets was larger than the mean for the captive ferrets in all 15 of these measurements. As with the males, basal length, coronoid height, and condyle height were all highly significantly different. Zygomatic width, rostral length, palatal length, and palatal width were also highly significantly different for the female analysis. As with the males, PC1 for females was driven by overall size, with all eigenvectors returning positive values. Unlike males, however, female captive and wild ferrets differed to a high significance (*p* < 0.001) for PC1—which accounts for 52.43% of the variation within the sample. The second and third principal components also separate wild and captive ferrets significantly. As with males, PC2, which accounts for 7.76% of female variation, and is mainly driven by post-orbital constriction, and condyle height inversely proportional to upper canine AP. The driving measures for PC3 of females were an inverse relationship between upper canine AP, and lower carnassial length relative to post-orbital constriction and bicoronion width.

### 3.2. Comparison of Black-Footed Ferret Groups

Differences arose when the ferrets were divided into their four groups (wild, early captive, recovery phase I, and recovery phase II) and compared for their mean scores for the cranial measurements via the step-wise discriminant analysis and for the principal component analyses. Of the 23 measurements, significant differences were found in the step-wise discriminant analysis for 9 and 10 variables for the males and females, respectively. The ferret groups and congeners differed in both sex-specific analyses in four measurements: Basal L, Max H, Bicoronion W, and Condyle H. Additionally, the male-only analysis discriminated among the groups in terms of Total H, Neurocranium W and L, Rostral L, and Bicanine W, while the female-only analysis discriminated among the groups in terms of Total L and W, Postorbital W, Upper Carnassial L, Palatal W and Temporal Fossa ML. In these analyses, the three taxa were distinguished from each other in all but one individual: one female steppe polecat was predicted to be a wild female black-footed ferret.

As expected, there was substantial overlap between the predicted and actual assignments within the black-footed ferret groups in the step-wise discriminant analyses, though, notably, the prediction errors only occurred (with three exceptions) between the two recovery phases and between the early captive and wild specimens (Table 5). In other words, except for three wild ferrets that were misclassified as recovery phase, all of the misclassifications were within the recovery groups (ten misclassifications) or with differentiating the early captive specimens from the wild ones (nine misclassifications). In fact, as many wild ferrets were predicted to be early captive specimens as were correctly identified as early captive (eight each). The majority of misclassifications across the entire ferret sample (19 of 22 specimens) were male while more than 95% of the female ferrets were correctly assigned. (Less than 77% of the ferret specimens were assigned to their correct groupings).

Only the first two principal components were able to distinguish between the four groups in the male black-footed ferrets in a Tukey–Kramer test. For PC1, both the wild group and recovery phase II group were significantly different from the recovery phase I group but not from each other. The second principal component could not distinguish the two recovery groups from each other but did distinguish both those groups from the early captive group and the wild group—which themselves could not be separated (Table 6).

In the female sample, there are significant differences between groups in the first four principal components. In PC1, recovery phase I is distinct from both recovery phase II and the wild group which were indistinguishable from each other (Table 6). For both PC2 and PC3, the wild group is statistically different from both recovery groups which were indistinguishable from one another. Finally, in PC4, the recovery phase II group significantly differs from the other two groups.

### 3.3. Comparison of Black-Footed Ferrets with Closely Related Species

Results from the multivariate discriminate analysis indicate that the two recent captivity groups are more distinct from the wild group than one of the outgroup comparison species is to the wild group on a number of measurements for males (Figure 3). There is overlap between Steppe polecats and both the wild and early captive group. The European polecats comprise the most distinct group sharing no overlap in morphospace with any other groups. In these axes, the modern captive groups are clearly more separated from the wild group than the Steppe polecat group with the recovery phase I group being slightly more different from the wilds than the recovery phase II group.

The female groups provide results that are slightly different than the results noted in the males (Figure 4). The biggest difference in the females as compared to the males is that there is no overlap between the 95% confidence ellipses of the Steppe polecat and the wild black-footed ferrets. The distance between the Steppe polecat group and the wild group is less than the distance between the wild group and the modern captive groups, particularly the recovery phase I group. Again, the European polecat group is extremely distinctive from the other groups. Additionally, the early captive group entirely encompasses the wild group, with the two groups sharing near identical positions on the canonical plot.

## 4. Discussion

### 4.1. Captivity and Cranial Morphology

The results support the main hypothesis that captivity drives changes in the cranial morphology of black-footed ferrets. This is demonstrated by the 12 and 15 of 23 measurements that were significantly different between captive and wild male and female ferrets, respectively. The most interesting revelation however, was not that the specimens were different but rather which cranial features were driving the difference. Research by Hartstone-Rose and colleagues [26] showed that in large felids, zygomatic width differed the most between captive and wild specimens. Anatomically, this variation in zygomatic width makes sense when considering mastication because it is the origin of the masseter muscle and the space under the arch restricts the passage of the temporalis muscle. Therefore, any changes in the development and size of the masseter and temporalis would likely directly affect the zygomatic arch [26,41]. Diets in captivity, though very similar to a natural diet in terms of nutrition, tend to lack the same structural components as a wild diet, usually being softer and less obdurate [21,26,27,41]. As a result, less force is needed to chew food and the mandibular adductors do not need to develop to the extent that they might in wild animals. The zygomatic width for females was significantly smaller in captive animals in our study, but did not have as strong an effect on dividing the wild and captive specimens as measures such as post-orbital constriction and condyle height—also both potentially related to the mandibular adductors with the post-orbital region being the anterior part of the temporalis origin and the coronoid process being the insertion of the temporalis muscle. This difference from the previous literature that we found in organizational emphases between captive and wild specimens is likely due to masticatory differences between felids and mustelids. As reported by Ewer [42], the masseter tends not to play as critical a role and the temporalis is overemphasized in mustelids relative to other carnivorans. Thus, cranial measurements regarding areas affected by these muscles will likely behave differently when comparing subsets of mustelid specimens as opposed to felid specimens.

While the narrowing of the postorbital constriction in the ferrets like the widening of the zygomatic arches of lions and tigers [26] in captive animals might be interpreted as allowing larger mandibular adductors though it should be noted that, because of differences in muscle architecture, larger muscles are not necessarily more forceful [43], the decreased condylar height found in captive ferrets compared to wild ferrets would be indicative of a loss of masticatory power; as condyle height decreases, it presents less surface area along the ramus and angle of the mandible for the attachment of the adductor muscles and also reduces their leverage [44] and it may indicate a lengthening of fascicles and reduction of pennation, both of which, in muscles of equivalent masses, would reduce their force capabilities.

Results including the outgroup species indicate that not only are there significant differences between the captive and wild black-footed ferrets, but that these differences are, in some ways, more pronounced than differences between congeners. It is important to note the relative closeness and even overlap (in the male specimens) of the wild ferrets and the Steppe polecats for the multivariate analysis (Figure 2 and Figure 3), especially when contrasted to the distance existing between the wild ferrets and the two modern captive ferret groups. This analysis demonstrates that the skulls of the wild and modern captive black-footed ferrets are actually more distinguishable than are the skulls of the wild black-footed ferrets and Steppe polecats—different species. These observations add perspective on how extreme the cranial changes are between captive and wild specimens. In this case, captivity can be seen as a greater driving force for variation in cranial morphology than speciation, results that are not only of importance for the black-footed ferret recovery program, but also in the broad fields of conservational biology and zoology.

### 4.2. Possible Effects of Diet on Cranial Morphology

While other aspects in the husbandry of the ferrets were changed as they were being prepared for reintroduction (e.g., transitioning them to more naturalistic enclosures), no other changes that could have affected their skull shapes were obvious [13,14].

Interestingly, our results directly contradict our second hypothesis that the captive specimens accessioned after the diet began changing to a less obdurate food source in 1996 would be less like the wild specimens than the captive specimens accessioned before 1996. The early recovery group significantly differed from the wild group in more measurements than the late recovery group significantly differed from the wild group. This hypothesis was developed around our previous finding that oral health was worse in the late recovery group compared to the early recovery group [21] and findings that poor oral health correlated to unnatural cranial morphology [27] (Figure 2). These findings raise the question of why the oral health of these specimen would worsen in the later recovery group but the cranial morphology would appear more natural in the same group? One likely answer is that the later specimens were fed a more mechanically natural diet but were also older at death. A more mechanically natural diet could explain the improved cranial morphology in the later group [26], and because oral health problems such as periodontal disease and calculus are known to progress with age in mammals [45] this combination of a more natural diet but increased age could explain the present findings. Unfortunately, the museum records available to us did not provide enough information to accurately determine the age at death of our specimens, and thus we cannot say definitively that the later accessioned specimens were older.

The only other factor that might have had a strong effect on the morphology of this perilously small genetic population might be the effects of inbreeding [30,46]. However, previous research found that inbreeding coefficients based on known pedigrees for captive specimens did not affect total skull length for ferrets after 1990 [14]. This is not to say that changes in genetic diversity did not contribute to changes in cranial measurements in the ferrets, but just that it is likely not the only, or even the strongest, driving factor.

The comparison of all four groups also revealed that the early captives and wild specimens did not differ significantly based on either cranial measurements or principal components, though they differ substantially in cranial anatomy from both of the recovery phase populations. This similarity among early captive and wild ferrets is most likely a result of wild ferrets being trapped and then accessioned as captive ferrets. As seen in the later captive breeding programs, breeding black-footed ferrets can be very difficult [18,47]. Therefore, because black-footed ferrets were plentiful during the early captive era, it was likely easier for captivity centers to trap wild ferrets than to breed captive ferrets. Thus, “early captive” ferrets were likely ferrets that spent a majority of their life in the wild but died in captivity.

A second possibility is that a lack of dietary standardization among early captives may have actually helped captive ferrets maintain natural cranial morphologic shape. A majority of these specimens were accessioned from 1880 to 1930, likely before captivity centers strongly regulated nutrition and feeding practices. As such, zoos may have fed ferrets a diet more consistent with the textures and mechanical properties of the ferrets’ natural diet—e.g., diets based on whole carcasses rather than the modern ground, nutritionally supplemented diets [13].

Another explanation for this phenomenon arises from the fact that most of the wild ferrets were accessioned around the same time as the early captive specimens—late 19th and early 20th centuries. It is possible that arriving at the brink of extinction simply brought about drastic physical changes to the species during the second half of the 20th century. By breeding the species from a single population of only seven reproductive individuals, evolutionary factors such as genetic drift or lack of genetic diversity would have been amplified, resulting in ferrets that were significantly different than their ancestors [30,46]. Not only time, but location may have driven this difference. The modern captive specimens were all descended from a single population from Wyoming, whereas the wild black-footed ferrets in our sample included specimens from a broader range where changes in availability of prey, presence of predators, and geography could have been causing different characteristics in cranial morphology to arise [48]. Therefore, we would expect to see similarities between the wild ferrets and the early captives, but not between the wild ferrets and the recovery phase ferrets.

Another factor that might have influenced the studied morphology is the effects that general stress in captivity can have on animal health, including cranial morphology. Indeed, many studies have documented indicators of stress that are more prevalent in captive than in wild animals [49,50] and that this increased level of stress can negatively impact the overall health of captive animals [51,52,53]. Thus, the main hypothesis of this study was simply that a difference between captive and wild specimens exists, and conclusively proving diet as a main cause of these morphological differences would likely require random assignment experimentation—something that clearly cannot be done in this rare taxon. With that said, based on the results of this study and supporting evidence from previous studies [26,54,55], we recognize other factors likely play a role in varying cranial morphology, but still consider diet as the primary cause for differences among captive and wild specimens.

### 4.3. Limitations and Future Directions

Clearly this study is limited by the highly restricted sample of specimens available of this critically endangered taxon. While this small sample size is particularly evident from the early captive population, it is important to emphasize that the original founder population was exceedingly small and thus the few individuals included in our sample represent a substantial proportion of that total population. With that said, the imbalances of the bins render many statistical analyses more speculative than truly descriptive of a larger theoretical population. Rather than statistically correcting for these imbalances, we chose to show the morphospace occupied by all individuals (e.g., in Figure 3 and Figure 4) to demonstrate the breadth of this morphological intraspecific diversity—especially relative to that found between species. Unfortunately, these small sample sizes render our analyses potentially more valuable qualitatively than quantitatively, but we do believe that they are still informative about the phenomena explored.

While this current study has provided valuable information on this important species, continued research would not only further benefit the black-footed ferret, but could also increase our understanding of the influence of captivity on the morphology of other mammalian species as well. One line of future studies would look into the exact mechanisms of mastication in these animals as well as other behaviors that could be affecting cranial morphology in captive specimens. For example, personal observations from John Ososky (of the Smithsonian), someone who has worked intimately with these animals, found that male ferrets were prone to gnawing on enclosures, and so further observations of these actions and possible other actions seen in captive specimens not typical of wild specimens could present further explanations for the variation noted between captive and wild specimens.

Previous research has found that the prevalence of periodontal disease in black-footed ferrets may also be affected by diet [21]. A study comparing changes in oral health to changes in cranial morphology could help not only to increase the understanding of both these metrics, but could even further deduce the problems with a mechanically unnatural diet in these animals.

As previously discussed, one of the difficulties of the current study was the lack of modern wild specimens. It may someday be possible to recover a large enough sample of reintroduced or wild born animals to compare to the present findings. As modern specimens are all presumably descended from former captive lines, we could use this population to better separate the morphological effects of diet and genetics in this species.

## 5. Conclusions

The current study furthered the findings of Wisely and colleagues [14] by documenting cranial changes from captivity in the black-footed ferret, with the inclusion of 16 new cranial measurements as well as comparative out-species groups to improve the knowledge of exactly in what ways these skulls were varying. In addition, captive ferrets were further divided into subgroups based on dietary phases that allowed us to observe the effects each specific phase had on black-footed ferrets and if there were positive effects to replacing the original “60/40” diet. Lastly, this study demonstrated that the crania of captive ferrets, in some ways, differ more greatly from wild ferrets than from those of two of their closest relatives. In other words, in some respects, captivity has a significantly greater effect on skull shape than does phylogenetic speciation—a finding that surely should be considered when managing captive populations of animals.

## Figures and Tables

**Figure 1 animals-12-02708-f001:**
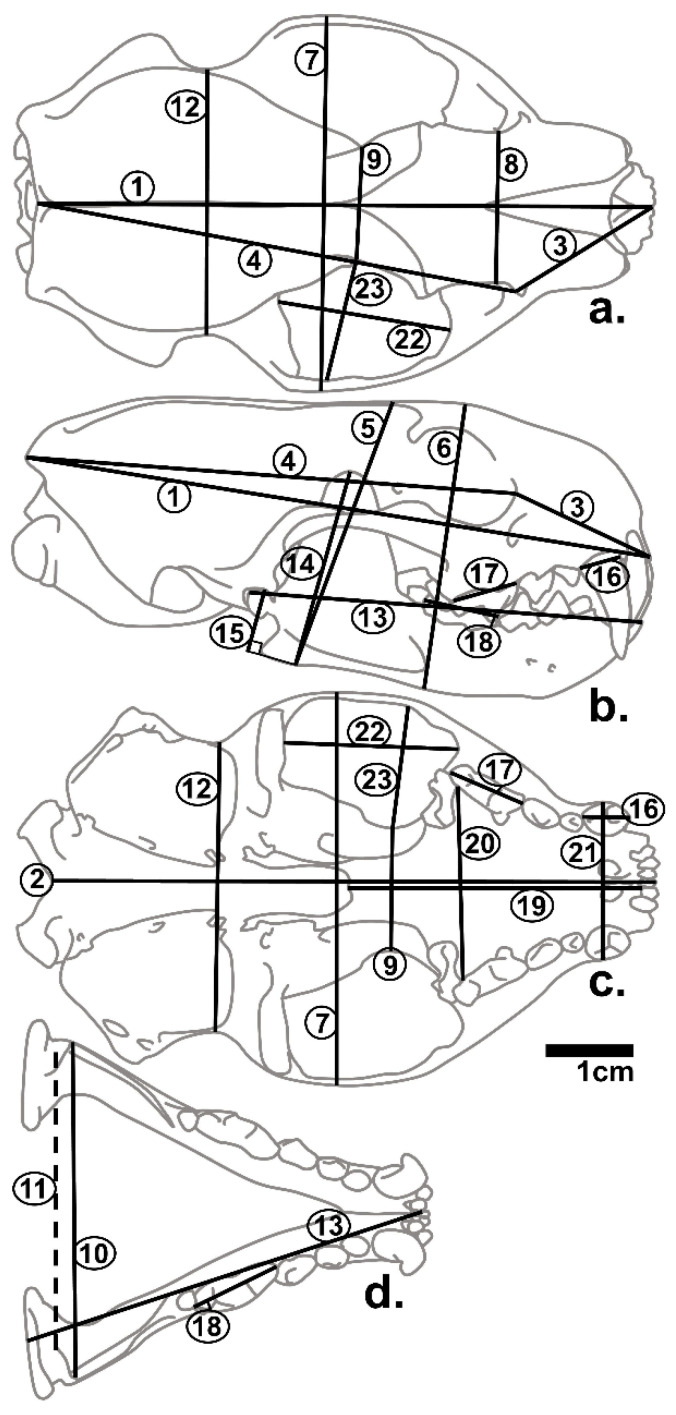
Measurements taken. Schematic view of *M. nigripes* skull in dorsal (**a**), lateral (**b**) and occlusal views (cranium, (**c**); mandible (**d**)). Rostral is to the right. Numbers correspond to descriptions Table 1.

**Figure 2 animals-12-02708-f002:**
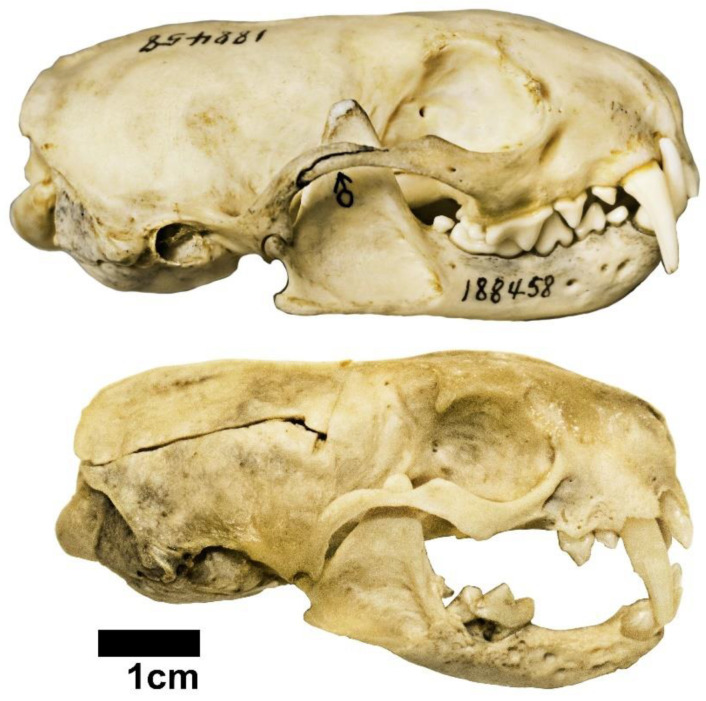
Example of typical wild (top; USNM 188458) and captive (SB 57; recovery phase 1) black-footed ferret skulls. Note poor oral health and cranial deformation (e.g., thin and distorted zygomatic arch) of this captive specimen.

**Figure 3 animals-12-02708-f003:**
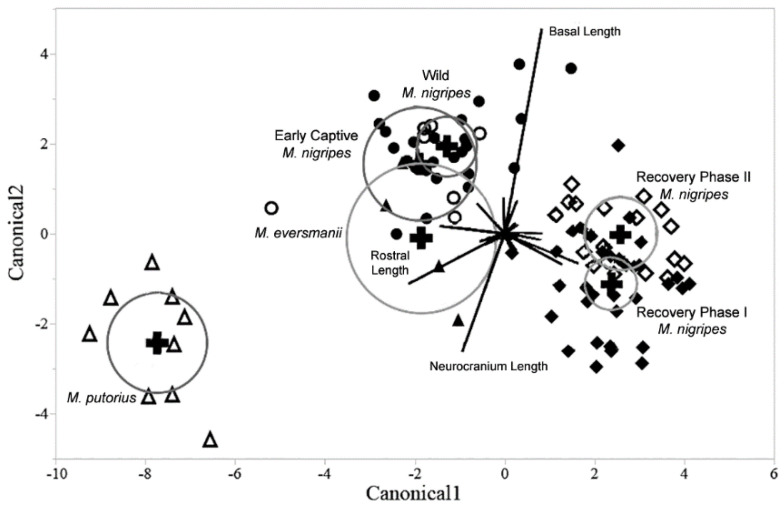
Canonical plot of discriminant analysis (with 50% density eclipses) for males for all four black-footed ferret groups and the two comparative outgroup species European and Steppe polecats (*Mustela putorius* and *M. eversmanii*, respectively). Wild group = filled circle; early captive = empty circle; recovery phase I = filled diamond; recovery phase II = empty diamond; *M. eversmanii* = filled triangle; *M. putorius* = empty triangle.

**Figure 4 animals-12-02708-f004:**
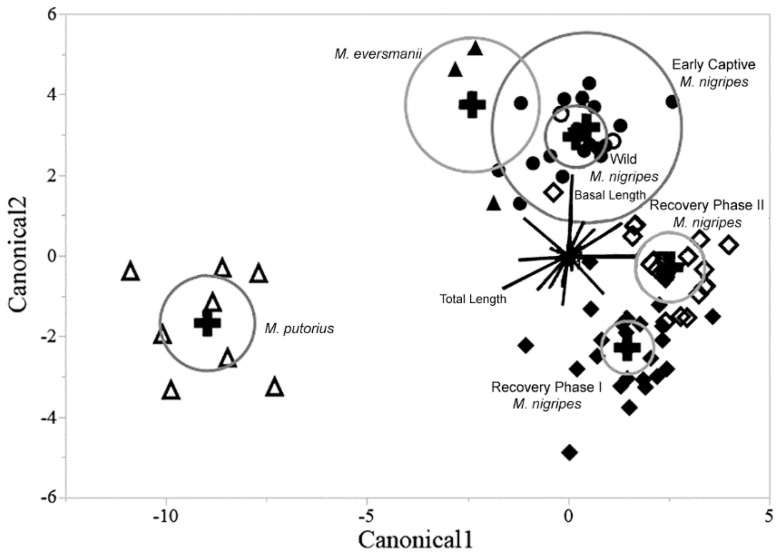
Canonical plot of discriminant analysis (with 50% density eclipses) for females for all four black-footed ferret groups and the two comparative outgroup species European and Steppe polecats (*Mustela putorius* and *M. eversmanii*, respectively). Markers are the same as in Figure 3.

**Table 1 animals-12-02708-t001:** Sample specimen distribution of the four black-footed ferret (*Mustela nigripes*) groups and the *M. putorious* and *M. eversmanii* specimens.

Group	Description	Male	Female	Total
Wild	Wild specimens accessioned prior to the modern reintroduction campaign (1876–1971)	30	20	50
Early captive	Specimens that were captive prior to the Black-Footed Ferret Recovery Act	8	3	11
Recovery phase I	Captive bred specimens fed the “60/40” diet from 1985 to 1996	53	58	111
Recovery phase II	Captive bred specimens from 1996 until the most recent accessions which were fed a less obdurate diet	18	16	34
*M. putorious*	European polecat	13	11	24
*M. eversmanii*	Steppe polecat	7	5	12
Total		129	113	242

Sex and group determined by museum records.

**Table 2 animals-12-02708-t002:** Description of craniomandibular measurements by bony landmarks.

Num. ^a^	Cranial Measurement	Description ^a^
1	Total length	Inion to alveolare (prosthion)
2	Basal length	Opisthion (inferior foramen magnum) to alveolare
3	Rostral length	Orbitale to alveolare
4	Neurocranium length	Orbitale to inion
5	Total height	Inferior mandibular angle to vertex
6	Max height	Greatest dorsal-ventral height
7	Zygomatic width	Zygonion to zygonion
8	Interorbital constriction	Supraorbitale to supraorbitale
9	Postorbital constriction	Cranial width at narrowest location posterior to orbits
10	Bicoronion width	Coronion to coronion
11	Biangular width	Gonion to gonion
12	Neurocranium width	Porion to porion
13	Jaw length	Interdentale to condylare
14	Coronoid height	Inferior mandibular angle to coronion
15	Condyle height	Inferior mandibular angle to dorsal-most condyle
16	Upper canine AP	Anteroposterior length of upper canine
17	Upper carnassial length	Anteroposterior length of upper carnassial
18	Lower carnassial length	Anteroposterior length of lower carnassial
19	Palatal length	Orale to staphylion
20	Palatal width	Endomolare to endomolare
21	Bicanine width	Maximum distance measured on lateral-most points on the upper canines
22	Temporal fossa AP	Anteroposterior length of the temporal fossa
23	Temporal fossa ML	Mediolateral width of the temporal fossa

^a^ Corresponds to Figure 1. All unilateral measurements measured on Right.

**Table 3 animals-12-02708-t003:** Comparative *Z*-tests of male captive (N = 85) and wild (N = 30) black-footed ferrets for each of the 23 cranial measurements and PC1–5.

Measurement	Captive $\bar{X}$	Captive SD	Wild $\bar{X}$	Wild SD	*p*-Value ^a^
Total length	64.58	0.2884	64.9	0.4284	0.5452
Basal length	62.32	0.2548	64.26	0.4058	<0.001 ***
Total height	27.69	0.1413	28.35	0.2302	0.0157 *
Max height	31.67	0.1411	30.97	0.2299	0.0112 *
Zygomatic width	40.69	0.2277	41.55	0.3919	0.061
Interorbital constriction	17.19	0.1074	17.47	0.1751	0.1756
Post-orbital constriction	11.81	0.1293	12.6	0.2106	0.002 **
Bicoronion width	30.76	0.1718	29.97	0.2864	0.0195 *
Biangular width	30.93	0.1852	31.25	0.3032	0.365
Neurocranium width	28.68	0.1677	28.83	0.2745	0.646
Rostral length	17.85	0.0991	18.28	0.1604	0.0252 *
Neurocranium length	51.43	0.2702	51.07	0.4014	0.4498
Jaw length	42.38	0.2332	42.98	0.3816	0.1807
Coronoid height	19.72	0.1147	20.69	0.1902	<0.001 ***
Condyle height	6.17	0.0724	7.37	0.1179	<0.001 ***
Upper canine anteroposterior	4.26	0.0301	4.25	0.0485	0.8747
Upper carnassial length	7.41	0.0374	7.44	0.0601	0.6421
Lower carnassial length	8.15	0.0483	8.35	0.0787	0.0373 *
Palatal length	31.9	0.1382	32.74	0.2238	0.0019 **
Palatal width	24.06	0.2106	24.99	0.3431	0.0231 *
Bicanine width	16.28	0.1106	16.74	0.1767	0.0311 *
Temporal fossa anteroposterior	17.93	0.1294	17.68	0.2146	0.3269
Temporal fossa mediolateral	16.8	0.1256	17.12	0.2121	0.1944
Principle component 1	−0.09	0.4385	0.85	0.6622	0.2404
Principle component 2	−0.5	0.1531	1.38	0.2312	<0.001 ***
Principle component 3	0.15	0.1529	−0.13	0.2308	0.3101
Principle component 4	0.11	0.1348	−0.104	0.2035	0.3851
Principle component 5	0	0.1314	−0.25	0.1984	0.2918

^a^ * = *p* < 0.05, *** = *p* < 0.01, *** = *p* < 0.001.

**Table 4 animals-12-02708-t004:** Comparative *Z*-tests of female captive (N = 79) and wild (N = 20) black-footed ferrets for each of the 23 cranial measurements and PC1–5.

Measurement	Captive $\bar{X}$	Captive SD	Wild $\bar{X}$	Wild SD	*p*-Value ^a^
Total length	59.89	0.267	60.85	0.425	0.0607
Basal length	58.3	0.221	60.59	0.421	<0.001 ***
Total height	25.59	0.14	26.34	0.272	0.0161 *
Max height	29	0.144	29.11	0.28	0.7277
Zygomatic width	37.42	0.178	39.4	0.347	<0.001 ***
Interorbital constriction	15.82	0.096	16.32	0.187	0.0199 *
Post-orbital constriction	11.33	0.121	12.1	0.234	0.0045 **
Bicoronion width	28.61	0.141	28.7	0.272	0.7601
Biangular width	29.05	0.179	29.71	0.345	0.0922
Neurocranium width	26.96	0.15	27.11	0.286	0.6574
Rostral length	16.63	0.089	17.38	0.173	<0.001 ***
Neurocranium length	48.07	0.274	48.53	0.436	0.3728
Jaw length	39.1	0.164	40.04	0.32	0.0103 *
Coronoid height	17.89	0.098	19.44	0.191	<0.001 ***
Condyle height	5.54	0.051	6.91	0.099	<0.001 ***
Upper canine anteroposterior	3.75	0.032	3.86	0.063	0.1282
Upper carnassial length	7.03	0.034	7.189	0.064	0.0352 *
Lower carnassial length	7.53	0.042	7.78	0.08	0.0069 **
Palatal length	29.61	0.131	30.63	0.255	<0.001 ***
Palatal width	22.61	0.209	24.48	0.407	<0.001 ***
Bicanine width	14.78	0.08	15.24	0.154	0.009 **
Temporal fossa anteroposterior	16.32	0.109	16.34	0.212	0.9075
Temporal fossa mediolateral	15.4	0.104	16.12	0.202	0.0019 *
Principle component 1	−0.52	0.465	2.51	0.7	<0.001 ***
Principle component 2	−0.15	0.162	1.44	0.244	<0.001 ***
Principle component 3	−0.17	0.172	0.88	0.259	0.0012 **
Principle component 4	−0.3	0.153	−0.07	0.23	0.4129
Principle component 5	0.05	0.147	−0.02	0.221	0.782

^a^ * = *p* < 0.05, *** = *p* < 0.01, *** = *p* < 0.001.

**Table 5 animals-12-02708-t005:** Discriminant analysis classification table (males, females; by group). Although the polecats were included in each sex-specific step-wise discriminant analysis, because all but one classified correctly (see text), they are omitted from this table for clarity.

		Predicted			
		Early Captive	Recovery Phase I	Recovery Phase II	Wild
Actual	Early Captive	6, 2	0	0	1, 0
	Recovery phase I	0	25, 24	6, 2	2, 0
	Recovery phase II	0	2, 0	15, 14	0, 1
	Wild	8, 0	0	0	17, 19

**Table 6 animals-12-02708-t006:** Connecting letters report for an All-pairs, Tukey comparative means test in the black-footed ferret groups first four principal components. Groups that do not share a common letter are significantly separated. Further PCs do not discriminate between groups.

Sex	PC	Early Captive *	Recovery Phase I	Recovery Phase II	Wild
Males	PC1	A B	B	A	A
	PC2	A	B	B	A
	PC3	A	A	A	A
	PC4	A	A	A	A
Females	PC1		A	B	B
	PC2		A	A	B
	PC3		A	A	B
	PC4		A	B	A

* Due to the female early captive small sample size, it was omitted from this analysis.

## Data Availability

Data are freely available on our website.

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
