# Peer review of "The Cranial Morphology of the Black-Footed Ferret: A Comparison of Wild and Captive Specimens"

_animals, 2022, doi:10.3390/ani12192708_

Round 1

Reviewer 1 Report

This manuscript is interesting however I have found some flaws that should be addressed before the final publication of manuscript.

Authors did not provide simple summary of the manuscript. Simple summary can provide much more information for an ordinary audiences. Abstract gave a lot of information regarding the background of research; authors are advised to focus on results and conclusion. The introduction is too long to understand, authors are advised to shorten it 

Author Response

Thank you for this valuable feedback.  We have cut the manuscript back substantially and added all of the sections required by the journal.

Reviewer 2 Report

Please find the attached file for detailed comments.

Author Response

See attached response

Reviewer 3 Report

Dear Authors

The manuscript submitted for review is interesting, but needs thorough improvement and is not suitable for publication in its current form. The authors have not followed the editorial requirements in the journal Animals, see: https://www.mdpi.com/journal/animals/instructions. Simple Summary is missing, literature citations do not comply with journal requirements. The References section is completely to be improved. The Introduction section is too long, it should be shortened. In this section, the authors refer to the characteristics of the masticatory muscles. I would ask , especially this part of the manuscript to be improved (page 3 and 4), because for anatomists it is too laconically written , and for non-specialists , for better understanding, a diagram of the muscles themselves and the initial and final attachments of the muscle units discussed would be required. Improve the quality and readability of Figures 2 and 3. In the supplementary material to the manuscript, include information on the statistical analyses performed (stepwise discriminant analysis as well as principal component analysis): for PCA (eigenvalue, % of total variance, cumulative eigenvalue, cumulative %), for DA (Wilks lambda, Wilks partial lambda, F deletions, P level, Tolerance, 1-Tolerance (R-square).

Regards

Author Response

Thank you for this valuable feedback.  We believe that we made all of these improvements and hope that you now find the manuscript suitable.

Round 2

Reviewer 3 Report

Thank you for incorporating some of the suggested improvements into your manuscript.